# Silencing Dicer-Like Genes Reduces Virulence and sRNA Generation in *Penicillium italicum,* the Cause of Citrus Blue Mold

**DOI:** 10.3390/cells9020363

**Published:** 2020-02-04

**Authors:** Chunxiao Yin, Hong Zhu, Yueming Jiang, Yang Shan, Liang Gong

**Affiliations:** 1Long Ping Branch, Graduate School of Hunan University, Changsha 410125, China; chunxiaoyin@scbg.ac.cn; 2Key Laboratory of Plant Resource Conservation and Sustainable Utilization, Key Laboratory of Post-Harvest Handling of Fruits, Ministry of Agriculture, Guangdong Provincial Key Laboratory of Applied Botany, South China Botanical Garden, Chinese Academy of Sciences, Guangzhou 510650, China; zhuhong@scbg.ac.cn (H.Z.); ymjiang@scbg.ac.cn (Y.J.); 3Core Botanical Gardens, Chinese Academy of Sciences, Guangzhou 510650, China

**Keywords:** citrus, *Penicillium italicum*, pathogenicity, RNA interference, cross-kingdom RNAi, Dicer, small RNA

## Abstract

The Dicer protein is one of the most important components of RNAi machinery because it regulates the production of small RNAs (sRNAs) in eukaryotes. Here, Dicer1-like gene (Pit-DCL1) and Dicer2-like gene (Pit-DCL2) RNAi transformants were generated via pSilent-1 in *Penicillium italicum* (Pit), which is the causal agent of citrus blue mold. Neither transformant showed a change in mycelial growth or sporulation ability, but the pathogenicity of the Pit-DCL2 RNAi transformant to citrus fruits was severely impaired, compared to that of the Pit-DCL1 RNAi transformant and the wild type. We further developed a citrus wound-mediated RNAi approach with a double-stranded fragment of Pit-DCL2 generated in vitro, which achieved an efficiency in reducing *Pi-Dcl2* expression and virulence that was similar to that of protoplast-mediated RNAi in *P. italicum,* suggesting that this approach is promising in the exogenous application of dsRNA to control pathogens on the surface of citrus fruits. In addition, sRNA sequencing revealed a total of 69.88 million potential sRNAs and 12 novel microRNA-like small RNAs (milRNAs), four of which have been predicated on target innate immunity or biotic stress-related genes in Valencia orange. These data suggest that both the Pit-DCL1 and Pit-DCL2 RNAi transformants severely disrupted the biogenesis of the potential milRNAs, which was further confirmed for some milRNAs by qRT-PCR or Northern blot analysis. These data suggest the sRNAs in *P. italicum* that may be involved in a molecular virulence mechanism termed cross-kingdom RNAi (ck-RNAi) by trafficking sRNA from *P. italicum* to citrus fruits.

## 1. Introduction

Citrus fruit production exceeded 140 million tons in 2016, accounting for one-fifth of the total fruit production worldwide [1]. After harvest, citrus fruits are commercially handled in packing houses in order to maintain their postharvest quality, to increase their shelf life, as well as to reduce losses due to pathogen infection. Citrus blue mold, caused by *Penicillium italicum*, is one of the most destructive pathogens of postharvest citrus fruits, resulting in losses of up to 80% [2].

Advancing our knowledge regarding the molecular mechanisms underlying interactions between citrus fruit and *P. italicum* is profitable to reduce postharvest losses. Regulating the ambient pH of the host is considered to be one of the pathogenic mechanisms of *Penicillium* species. These pathogens can produce organic acids and deplete ammonium to acidify the ambient environment of citrus fruits during decay development, which is a cue for virulence enhancement [3]. In addition, suppressing host defenses has been considered another available pathogenic mechanism of *P. digitatum*. Macarisin et al. demonstrated that *P. digitatum* can suppress the host’s production of reactive oxygen species, while the non-host pathogen *P. expansum* can trigger a 63-fold increase in hydrogen peroxide production in citrus fruits [4]. In addition, comparative genomics has been developed to analyze the molecular basis of fungal pathogenicity, and a large number of putative virulence factors, including Carbohydrate-Active enZymes, proteases, and oxidoreductases, were identified in *Penicillium* species, these genes were up-regulaed during the infection process, which is highly related to their pathogenicity [5,6]. Recently, cross-kingdom RNA (Ribonucleic Acid) interference (ck-RNAi) mediated by small RNAs (sRNAs) has been recognized as an innovative molecular mechanism to modulate fungal virulence [7]. However, the presence and function of sRNAs in *P. italicum* has never been elucidated.

Dicer, an RNase III-like endonuclease that is a major component of RNAi machinery, processes long-chain double-stranded RNA (dsRNA) into 21–25 nt-long small interfering RNAs (siRNAs) [8]. These siRNAs are integrated with the Argonaute (AGO) family proteins to form the RNA-induced silencing complex (RISC), guiding the degradation or translational inhibition of the target mRNA [9]. It is well-known that the RNAi pathway in filamentous fungi is associated with the dicer-dependent process in the generation of siRNAs, including microRNAs (miRNAs) and small interfering RNAs (siRNAs) [10]. In addition, Dicer-like genes play important roles in fungal conidiation and pathogenicity; for example, Meng et al. (2017) reported that a Dicer-like gene 2 mutant of the entomopathogenic fungus *Metarhizium robertsii* exhibited a 55.8% reduction in conidial yield [11]. In the plant pathogenic fungus *Fusarium graminearum*, ascospore discharge was decreased in the Dicer-like gene 1 deletion mutant and was completely blocked in the Dicer-like gene 1 and 2 double deletion mutant [12]. Dicer-like proteins regulate growth and conidiation in *Colletotrichum gloeosporioides*, and the double mutation abrogated the ability to penetrate *Hevea brasiliensis* [13]. These findings suggest that filamentous fungi employ RNAi-related pathways in various physiological processes to adapt to different environmental conditions; however, the RNAi machinery has never been reported in *P. italicum,* although it is the most serious pathogen causing the decay of citrus fruits at the postharvest stage.

sRNAs were first discovered in the model fungus *Neurospora crassa* in 2010 [14]. Since then, they have been widely reported in other filamentous fungi, such as *Trichophyton rubrum* [15], *Sclerotinia sclerotiorum* [16], *Fusarium oxysporum* [17], *Blumeria graminis* [18], and *Valsa mali* [19]. However, fungal sRNAs remain functionally uncharacterized, although they play crucial roles in animals and plants during diverse developmental processes under both normal and stress conditions [20]. However, the rapid development of sRNA sequencing technology provides a novel strategy to detect and identify precursors or mature sequences of sRNA in fungi [21] and thus can be a useful tool to identify sRNAs in filamentous fungi.

The aim of this study was to investigate the function of Dicer-like genes in relation to the pathogenicity of *P. italicum* by using protoplast and citrus wound-mediated RNAi technology, in addition to discovering and characterizing the sRNAs in the RNAi transformants by using the Illumina sequencing platform. Our study revealed the potential linkage between the RNAi machinery and the pathogenicity of *P. italicum*, and highlighted the probability of utilizing RNAi as a tool to control *P. italicum* infection in postharvest citrus fruits.

## 2. Materials and Methods

### 2.1. Fungal Strain and Plasmid

*P. italicum* was isolated from a decayed citrus fruit, as reported in a previous study [22]. This strain was maintained on potato dextrose agar (PDA) containing 0.4% potato extract, 2% dextrose, and 2% agar at 25 ± 2 °C for 5 days. The spores from a 5 day-old culture were collected, filtered, and adjusted to a suitable concentration using a hematocytometer and were used in the following experiments. The pSilent-1 plasmid containing the hygromycin resistance gene as a fungal selection marker, and the promoter and terminator of the trpC gene from *Aspergillus nidulans* [23] was kindly provided by Pro. QB Hu (College of Agriculture, South China Agricultural University, Guangzhou, China).

### 2.2. Construction of Silencing Vectors

Two Dicer-like genes were annotated from our sequenced *P. italicum* genome (GCA_002116305.1) and released in GenBank under accession numbers MN537145 and MN537146. The partial CDS sequences of Dicer 1-like gene *DCL1* (582 bp) and Dicer 2-like gene *DCL2* (406 bp) were separately amplified using gene-specific primers (Appendix A). The silencing vector was constructed using pSilent-1 by inserting the sense sequence of the partial CDS at the XhoI/HindIII sites and the reverse complement sequence of the partial CDS at the KpnI/ApaI sites, according to Chen et al. [23]. The prepared silencing vectors were further confirmed by DNA sequencing at IGE Biotechnology (Guangzhou, China). The empty pSilent-1 plasmid was used as a negative control.

### 2.3. Preparation of Protoplasts, Transformation and Screening

Protoplasts were prepared, as described by Visser et al. [24], with minor modifications. Briefly, conidial suspensions (10^7^ spores/mL) were cultured in potato-dextrose broth (PDB) containing 0.4% potato extract and 2% dextrose with shaking at 200 rpm for 48 h at 28 °C. Then, the mycelia were collected and washed with a 0.8 M NaCl solution. A liquid enzyme mixture consisting of driselase (Solarbio, Beijing, China), cellulase (Solarbio, Beijing, China), and sterile d-glucose solution (Sigma-Aldrich, Guangzhou, China) was used to fully digest fungal cell walls. The obtained protoplasts were further examined with an optical microscope to ensure complete digestion of the cell walls.

Transformation was performed, as described by Proctor et al. [25], with the following modifications. The constructed pSilent-1 plasmid (10 µg, 800ng/µL) was mixed with 100 µL of the protoplasts solution prepared, as described above (10^8^ cells/mL), and incubated at room temperature for 10 min; 200, 400, and 800 µL of 50% PEG 4000 solution (50 mM Tris-HCl (pH 8.0) and 50 mM CaCl_2_) was then gradually added and mixed thoroughly with the protoplast solution. After 30 min at room temperature, the supernatant was removed by centrifugation (4 °C, 3000× *g*, 5 min), and the protoplasts were resuspended in 1 mL of STC solution (50 mM CaCl_2,_ 50 mM Tris HCl (pH 8.0), 0.8 M sorbitol). PDA (5 mL, with the addition of 0.6 M glucose) was added and gently mixed with the protoplast solution. The mixtures were poured evenly on PDA plates (0.6 M glucose, 200 µg/mL hygromycin B) and incubated at 28 °C for 48 h until transformant mycelia were visible. The transformants were subcultured alternately on PDA plates containing hygromycin B (300 µg/mL) or not containing hygromycin B three times.

The wild-type strain and the transformants were cultured separately in PDB medium with shaking (200 rpm) for three days. Genomic DNA was extracted by using a HiPure Fungal DNA Mini Kit (Magen, Shanghai, China). Screening was performed by amplifying the hygromycin resistance gene (hph) with specific primers (Appendix A). The amplification reaction contained 200 ng of DNA, 0.5 μM hph forward and reverse primers, 25 µL of 2× reaction polymerase buffer mix (TSINGKE, Beijing, China), and sterile water was added to a total volume of 50 µL. Then, the following PCR (Polymerase Chain Reaction) conditions were used: 94 °C for 4 min as the initial denaturation step; followed by 35 cycles at 94 °C for 30 s for denaturation, 60 °C for 30 s for primer annealing, and 72 °C for 1 min for DNA strand synthesis; and a final extension step at 72 °C for 10 min. Amplified products were stained with EtBR (Sigma, Saint Louis, MO, USA) in 1.0% agarose gel electrophoresis and visualized by GelDoc-It 310 Imaging System. (Thermo Fisher Scientific, Boston, MA, USA).

### 2.4. Biomass Assay and Pathogenicity Testing

To compare the biomass of the mycelia and spores of the prepared strains, 100 µL of a conidial suspension (10^7^ spores/mL) of the transformants and the wild-type strain were inoculated into 50 mL of PD liquid medium in a 100 mL conical flask and placed on a shaker at 200 rpm for 3 days at 28 °C. The mycelium was collected by vacuum extraction with a Buchner funnel and filter paper; after freeze drying under a vacuum for 12 h, the mycelium was weighed. Conidiospores were harvested by filtering (0.4 µm) and were further counted under an optical microscope with a hemocytometer. Each treatment was repeated three times, and four parallel experiments were performed for each treatment.

Inoculation of the *P. italicum* strains on sugar oranges, which is a Chinese mandarin with a special feature of granulated epidermis pattern and is a local citrus fruit of Guang dong province, was performed according to our reports [22]. Briefly, the matured citrus fruits with the diameter of around 6 cm were chosen and cleaned with sterile water, and then dabbed with 70% ethanol within 10 s. After air drying, small wounds (5 mm deep and 2 mm wide) were created by a sterile knife tip at two positions opposite one another near the equator of the fruits. Conidial suspensions (10 µL, 10^6^ spores/mL) from the transformants and the wild-type strain were inoculated separately on the surface of the wounds, and the citrus fruits were then kept in a chamber at 28 °C for 5 days. Thirty citrus fruits were used for each treatment with two independent biological repeats. On days 3, 4, and 5 after inoculation, the diameters of the infected sites were measured, and the samples of the infected epidermis were collected and stored at −80 °C for gene expression analysis.

### 2.5. Preparation of Pit-DCL2 dsRNA and Application on the Citrus Fruit Surface

The double-stranded RNA (dsRNA) of *Pit-DCL2* was synthesized by using a HiScribe kit (NEB, Beijing, China) following the manufacturer’s protocols. The specific primers Pit-DCL2-F and Pit-DCL2-R, both of which contained a T7 promoter sequence at the 5′ end (Appendix A), were designed for dsRNA amplification. dsGFP was used as the negative control, and the concentration of the prepared dsRNA was confirmed with a Nanodrop 2000 (Thermo Fisher Scientific, Boston, MA, USA). Two different methods were used to apply dsRNA to the surface of the citrus fruits: In the first method, two wounds opposite of each other on the equator of the sugar orange were created with a sterile knife, and 200 and 800 ng of dsDCL2 or 800 ng dsGFP was dropped onto the wound surfaces. After 2 h, 10 µL (10^5^ spores/mL) of conidial suspension was inoculated at the same positions on the wounds. In the second method, 200 and 800 ng of dsDCL2 or 800 ng of dsGFP were separately incubated with protoplasts (10^7^ cells/mL) in PTC solution at room temperature for 1 h, and the mixture (10 µL) was then inoculated on each wound. Four experimental groups were arranged, including the control group (without dsRNA), dsGFP (800 ng), dsDCL2 (200 ng), and dsDCL2 (800 ng). Ten sugar oranges were used for one parallel experiment and a total of thirty sugar oranges were tested for each experimental group, and the treated fruits were placed in an incubator at a constant temperature of 28 °C for 5 days. The decay index was calculated according to the formula: Σ(decay scale × proportion of fruits corresponding to each scale)/5n × 100% (*n* = 60) [22], and the infected citrus epidermises were collected and stored at −80 °C for gene expression analysis.

### 2.6. Small RNA Sequencing

Nine sRNA libraries (three DCL1RNAi, three DCL2RNAi and three wild-type) containing the total RNA with lengths of 18–30 nt were prepared by using TruSeq small RNA library kits (Illumina Trading, Shanghai, China). The libraries were barcoded and sequenced using the Illumina HiSeq^TM^ 2500 SE50 platform by Magigene Biotechnology Co. (Guangzhou, China). Raw sRNA-seq reads were first processed through custom Perl and python scripts to remove reads containing poly-N and adapter sequences and low-quality reads. In addition, the Q20, Q30, and GC content of the raw data were calculated. Then, reads with a length of 18–30 nt were selected from the clean reads for all downstream analyses. The total clean sRNAs were mapped to the *P. italicum* PHI-1 genome sequence (GenBank assembly accession number: GCA_000769765.1) by Bowtie (http://bowtie-bio.sourceforge.net/index.shtml) with a mismatch value of 0, and the expression and nucleotide length distribution, and the 5′ terminal nucleotide composition were analyzed with a set of Perl and R scripts. The mapped sRNAs were then aligned with the Rfam database (version 14.0) to remove sRNAs originating from rRNAs, tRNAs, snRNAs, and snoRNAs (http://rfam.xfam.org/). miRBase 20.0 (http://www.mirbase.org/) was used as a reference to find known miRNA. Reads overlapped with protein-coding genes and repeat sequences were analyzed with BEDTools (https://bedtools.readthedocs.io/en/latest/). Candidate miRNAs were predicted with miREvo [26] and mirdeep2 [27] using the default settings. The secondary structures of the miRNAs were predicted using RNAfold [28]. The sRNA targets were predicted by TargetFinder 1.6 [29] with the cDNA sequences of an orange species *Citrus sinensis* (www.citrusgenomedb.org/). All raw data generated in this study were deposited in the NCBI Sequence Read Archive database under the accession no. GSE138683.

### 2.7. qRT-PCR and Northern Blot Analysis

Quantitative real time PCR (qRT-PCR) analysis was performed using a 7500 Fast Real-Time PCR System (Applied Biosystems, Boston, MA, USA) with the following conditions: The in front of total RNA was extracted using the HiPure Mini Kit (Magen, Shanghai, China) according to the manufacturer’s specifications. cDNA was generated with PrimeScript^TM^ RT Master Mix (TaKaRa, Tokyo, Japan) following the manufacturer’s instructions. qRT-PCR was conducted using 20 ng of cDNA, 10 µL of 2× TB Green Premix Ex Taq^TM^ II (TaKaRa), 0.4 µL of forward primers and reverse primers (Appendix A), 0.4 µL of ROX Reference Dye II (TaKaRa), and the reaction volume was brought to a total volume of 20 µL with sterile water. The relative expression levels of target genes were first normalized to the endogenous reference gene by using the formula 2^−ΔΔT^ [30], then normalized relative to the level of gene transcripts in the control group, and the β-actin gene was used as the reference gene (Appendix A). All experiments were repeated three times.

For Northern blot analysis, 100 µg of total RNA was resolved on a 15% urea denaturing polyacrylamide gel and transferred to Amersham Hybond^TM^-NX membranes (GE Healthcare, USA). The 21–22-nt microRNA oligonucleotide probes (Appendix A) were labeled using a Biotin 3′ End DNA Labeling Kit (Thermo Fisher Scientific). The U6 probe was used as the control to confirm uniform loading. Membranes were hybridized for 16 h at 37 °C and washed three times at the same temperature with stringent washing buffer containing 1% SDS and 1× SSC. The membranes were blocked, washed, and chromogenically reacted with a Chemiluminescent Nucleic Acid Detection Module (Thermo Fisher Scientific); hybridization signals were then detected using a ChemiDoc^TM^ Touch Imaging System (Bio-Rad, Hercules, CA, USA).

## 3. Results

### 3.1. Generation of Silencing Transformants of Dicer-Like Genes in P. italicum

The silencing vectors expressing hairpin dsRNA of the DCL1- and DCL2-like gene fragments were constructed and transformed separately into *P. italicum*. As shown in Figure 1, the transformants were identified by the expected amplicon of 519 bp and the absence of this amplicon in the wild-type strain (Figure 1a). Spore production was not significantly different between the wild-type strain and the transformants, but the DCL1RNAi and DCL2RNAi transformants had slightly higher mycelial production than the blank vector control and the wild-type strain (Figure 1b,c). However, the spore and hyphal morphology exhibited no visible changes in the transformants (data not shown). In addition, the expression levels of the DCL1-like gene and DCL2-like gene in the transformants were significantly reduced by 5.8 and 2.4 fold, respectively, compared to those in the wild-type strain and by 5.8 and 3.0 fold, respectively, compared to those in the blank vector-transformed strain (Figure 1d), suggesting that we obtained RNAi transformants with downregulated DCL-like genes in *P. italicum*.

### 3.2. The Effect of Dicer-Like Gene Silencing on the Infection of Citrus Fruits by P. italicum

The pathogenicity of the DCL1RNAi and DCL2RNAi transformants was tested on sugar oranges. As shown in Figure 2, the pathogenicity of the DCL1RNAi transformant was not significantly different compared to that of the wild-type or blank plasmid-transformed strain, but the ability of the DCL2RNAi transformant to infect the citrus epidermis was significantly limited; however, the expression profile of the DCL-like genes in infected citrus tissues revealed that both Pit-Dcl1 and Pit-Dcl2 were significantly decreased after inoculation on day 4 and day 5 (Figure 3). These results suggested that Pit-DCL2 but not Pit-DCL1 participates in the pathogenicity of *P. italicum*. Furthermore, we performed the pathogenicity test again by using the protoplast and citrus peel-mediated RNAi methods, respectively, in which the prepared dsDCL2 construct was treated to the protoplast of *P. italicum* or citrus peel before the infection; as shown in Figure 4, although the method of protoplast-mediated RNAi seemed more effective in achieving an RNAi response, both methods produced the expected results of downregulating the *Pit-DCL2* gene and weakening the pathogenicity of *P. italicum.* However, this efficiency was dose-dependent, suggesting that higher concentrations of the treated dsRNA could achieve higher gene silencing efficiency.

### 3.3. Characterization of Small RNAs in the RNAi Transformants

We sequenced nine sRNA libraries prepared from the DCL1RNAi and DCL2RNAi transformants as well as the wild-type strain by using an Illumina platform. A total number of raw reads (115.77 millions), ranging from 10.89 to 15.41 million reads, were generated from the nine individual libraries (Appendix A). After adapter removal and quality filtering, a total of 110.77 million clean reads were used for further analysis; the clean reads with lengths of 18–35 nt were termed total sRNA (69.88 million). The total sRNA was observed to have 64.75–83.49% of reads mapping to the referenced *P. italicum* genome (Appendix A). The mapped total sRNA was further classified according to the origin from different genomic regions, and the classification suggested that the valid sRNA was mainly produced from noncoding RNA species, including rRNA, tRNA, and snRNA (Figure 5b). However, the percentage of the sRNA matching rRNA and exon sequences was greatly reduced in the DCL2RNAi and DCL1RNAi samples compared to that in the wild-type samples (Figure 5b). In addition, the size distribution of the sRNA showed a peak at 19–21 nt and 21–23 nt in both the wild-type and the DCL1RNAi mutant, but the peaks were shifted to 21–23 nt and 23–25 nt in the DCL2RNAi mutant (Figure 5a), and preferential accumulation of 5′-U (uracil) sRNA was found in both the wild-type strain and the RNAi mutant transformants (Figure 5c).

### 3.4. Structural Features, Expression Patterns and Predicted Targets of the Novel Pit-milRNAs

As there is no currently available miRNA database for fungal species, we used miRNA sequences published in miRBase database to search the known miRNAs in the sequenced libraries, which revealed 15 known miRNAs in *P. italicum* (Appendix A). However, we performed a prediction of the novel miRNAs in the sequenced libraries, and the results identified a total of 12 novel milRNAs (microRNA-like RNAs), among which Pit-novel 7 had the lowest MFE (minimum free energy), suggesting good structural stability. The stem-loop structures of these milRNAs were predicted by RNAfold (Appendix A). On average, the number of novel milRNAs was significantly reduced in the RNAi transformants compared to that of the wild-type strain; in addition, novel milRNAs (Pit-novel 6 and Pit-novel 7) were found in DCL1RNAi and the wild type, but not in DCL2RNAi (Figure 6a). Primers covering the complete precursor sequences of the Pit-milRNAs were designed to perform expression analysis via RT-qPCR. Five of the primer pairs worked well, and the results showed that the precursor sequences of miRNAs (Pit-novel 2, 6, 7 and 11) accumulated in the RNAi transformants (Figure 6b), suggesting that the cleavage ability of Dicer was weakened in the RNAi transformants. However, we only detected one of the milRNA candidates (Pit-novel 7) by Northern blotting, which showed an increased amount of the precursor sRNA and a decreased amount of the mature sRNA in the RNAi transformants compared to the wild-type strain (Figure 6c), consistent with the expected results. Nevertheless, we failed to detect other Pit-milRNAs, which may be caused by an insufficient abundance of the sRNAs under the culture conditions. Among these Pit-milRNAs, four were predicted to have five functionally annotated target genes in *C. sinensis,* and Pit-novel 7 was predicted to originate from a locus in contig1283 at the antisense gene link region between 115,664 and 115,928. Further, Pit-novel 7 milRNA was found to have one target in the *C. sinensis* genome, which was annotated as an AP2/B3-like transcription factor (Table 1).

## 4. Discussion

*P. italicum* is responsible for serious economic losses of citrus fruits during postharvest storage and transportation. Currently, the application of synthetic fungicides is the major method for controlling the postharvest citrus pathogens, but those fungicides have shown limited success. Therefore, an alternative strategy is required. In the field of life science, although RNAi technology is not the most efficient tool for the functional characterization of novel genes compared to approaches such as CRISPER-Cas9 technology, it has emerged as the alternative strategy with the greatest potential to control agricultural pests in a way that blocks gene expression [31]. However, this novel strategy is required to widely characterize pathogenesis-related genes in filamentous fungi [32]. Our findings demonstrated that the *Pit-DCL2* gene is a primary contributor to pathogenesis (Figure 2) and also showed the biosynthesis and size distribution of sRNAs in *P. italicum* (Figure 5). This result was consistent with previous reports on the plant pathogenic fungi *V. mali* [12] and *Colletotrichum gloeosporioides* [13], suggesting that this gene may be a potential target for the RNAi-based control of fungal diseases. Indeed, external spray application of dsRNA targeting fungal DCL-like genes significantly inhibited the development of gray mold disease in different fruits and vegetables [33].

The efficiency of dsRNA application on the surface of citrus fruits could be improved by increasing the amount of dsRNA. As shown in Figure 4, the application of 800 ng dsDCL2 had a higher efficiency than that of 200 ng dsDCL2 in citrus wound-medicated RNAi. This approach includes a key step of incubating dsRNA with citrus wounds, which achieved a gene silencing efficiency similar to that of incubating dsRNA with *P. italicum* protoplasts (Figure 4); doing this incubation before the infection is also a key step in protoplast-mediated RNAi. Indeed, protoplast-mediated RNAi without the barrier of the fungal cell wall has proven to be immensely successful in filamentous fungi [34,35]. It is accepted that exosomes (extracellular vesicles) play a central role in cell-to-cell communication, for example, Arabidopsis secretes exosome-like extracellular vesicles to deliver sRNA into *Botrytis cinerea* to silence virulence-related genes [7], and exosomes are involved in the transfer of small RNAs by nematode parasites to mammalian cells [36]. Citrus wound-derived vesicles might play key roles in transferring dsRNA into fungal cells, and if so, citrus exosomes are likely to be employed in exogenous dsRNA-mediated RNAi for the development of fungicides against *P. italicum*. Indeed, grapefruit-derived nanovesicles can transport siRNAs to different types of mammalian cells [37]. In contrast to siRNA amplification in fungi, siRNAs in *Caenorhabditis elegans* can efficiently develop into substantial amounts of siRNAs by promoting secondary siRNA amplification [38]. There is currently no evidence to show that fungi have the ability to maintain secondary siRNA amplification in vitro, although dsRNAs being dripped onto plant surfaces may induce the plant RNAi machinery to amplify fungal dsRNA. Our results are consistent with the silencing of *Myo5* in *Fusarium asiaticum* via spray-induced gene silencing, which shows an increased efficiency of dsRNA uptake via the wounded plant surface [39]. However, dsRNA surface treatment may be hard to protect non-wounded citrus against *P. italicum,* but we think that the composition of the dsRNA preparation may be improved by adding citrus exosomes.

Cross-kingdom RNAi (ck-RNAi) is considered to be a natural phenomenon by which sRNAs are transferred between host and pathogens and has not only provided significant new insights into the pathogenic mechanisms of plant pathogens, but has also opened a new avenue to generate environmentally friendly fungicides [40,41]. While natural ck-RNAi has attracted great interest in theoretical and applied studies underlying host-pathogen interactions and approaches for managing plant pathogens, surprisingly, only a few studies have been reported in the literature. There are two well-documented examples of natural ck-RNAi: *Botrytis cinerea* Bc-siR37 suppresses plant defense genes, including At-WRKY7, At-PMR6, and At-FEI2 [42], and *Puccinia striiformis* Pst-milR1 acts as an effector to suppress host immunity by binding the wheat *PR2* gene [43]. However, Kettles et al. recently confirmed the lack of evidence for ck-RNAi during the wheat-*Zymoseptoria tritici* interaction [44]. Currently, our understanding of natural ck-RNAi is still in the initial stage, and additional research studies will promote the development of this field. DCLs are the most important components of the RNAi machinery in eukaryotes, and in the present study, downregulation of the DCL2-like gene was revealed, but the DCL1-like gene in *P. italicum* did not significantly reduce in its ability to infect citrus fruits (Figure 2 and Figure 3). Subsequent sRNA sequencing also indicated that the DCL2-like gene plays a more important role in regulating milRNA expression than the DCL1-like gene in *P. italicum* (Figure 6)*,* as similarly reported in *Magnaporthe oryzae* [45]. These data suggest that *P. italicum* may possess a mechanism of RNAi or ck-RNAi and that its sRNA might be an effector in citrus epidermal cells that induces the RNAi response of citrus fruits during the infection process.

Furthermore, we predicted 12 novel sRNA precursors with a microRNA-like stem-loop secondary structure in the *P. italicum* sRNA libraries (Appendix A). In addition, a high-throughput in silico data analysis revealed 24 potential candidate milRNAs in *Penicillium marneffei* [46] and 34 in *Penicillium chrysogenum* [47]. Why did we identify a small amount of novel milRNAs in *P. italicum*? The reason may be that the collected *P. italicum* samples were not at different developmental stages or at the infectious stage in the citrus fruit. It has been reported that Pt-sRNA can be especially accumulated in *Penicillium triticina*-infected wheat leaf tissue [48] and that a number of *Blumeria graminis* sRNAs were significantly differentially expressed at 0 to 48 h after inoculation in barley [18]. These findings reflect that internal needs or environmental stresses might form a feedback regulation mechanism to stimulate the accumulation of sRNAs in fungi or oomycetes.

In fact, the number of sRNA candidates, in particular those with the potential for cross-kingdom transfer, is rather limited, so it is hard to overcome the huge number of plant immune-related genes. Moreover, we considered that only some sRNAs can travel across the boundaries between plant pathogens and their hosts to silence genes in the interacting plant hosts, while most of the remaining sRNAs function endogenously. However, we have to point out that most fungal sRNAs, although they do not contain a typical hairpin structure, may also be functional in inducing RNAi during plant-microbe interactions as well as during the process of their own growth and development. In filamentous fungi, the dicer-dependent RNAi process is recognized to generate two types of sRNA, namely, miRNA, and siRNA. Many studies have demonstrated that most sRNA should exist with structural features of siRNA. A total of 1–1.28 million potential sRNAs and only two candidate milRNAs were identified by using Illumina sequencing in *P. triticina* [48]. A total of 600–700 sRNAs were obtained from *Trichoderma reesei* by Solexa sequencing, but only 13 milRNAs were predicted based on hairpin structure analysis [49]. Indeed, dsRNA/siRNA from a vector expressing the transgene construct also worked well, with specific efficiency to silence related genes in fungi [34,35]. Consequently, we need to develop a pipeline to identify siRNA production and a reliable methodology to experimentally characterize functions of siRNAs in fungal species in the future.

Considering that natural ck-RNAi must show the evidence of the potential target genes in the host, we predicted the targets of the 12 novel Pit-milRNAs against the *Citrus sinensis* genome by using TargetFinder 1.6. Interestingly, we found that all of the putative target genes play critical roles in plant innate immunity or the biotic stress response (Table 1); for example, Pit-novel 1 targets a host gene encoding a leucine-rich repeat (LRR) receptor kinase, which is the largest subgroup of the receptor-like kinase family and functions as the first barrier in the plant innate immunity to recognize potential invaders [50]. Subtilisin-like proteases, targeted by Pit-novel 11, are also receptors associated with plant-pathogen recognition and immune priming [51]. These findings suggest that natural ck-RNAi is a potential virulence mechanism involved in interactions between citrus fruits and *P. italicum.* However, the direct interactions between Pit-milRNAs and their target host genes are unclear, and this knowledge will depend on technological advances in the future.

## 5. Conclusions

Understanding the molecular mechanism of ck-RNAi will help us to develop innovative strategies for crop protection. This purpose motivates us to explore the RNAi machinery and the biogenesis and function of sRNA in *P. italicum.* As revealed in this study, Pit-DCL2-mediated sRNA may play an important role in *P. italicum* pathogenicity. In addition, a method for direct exogenous application of dsDCL2 molecules was developed, based on citrus wound-mediated RNAi and showed great potential to trigger RNAi in *P. italicum.* Hence, for the first time, we illustrated that exogenous dsRNA application can be used to control diseases in postharvest citrus fruits.

## Figures and Tables

**Figure 1 cells-09-00363-f001:**
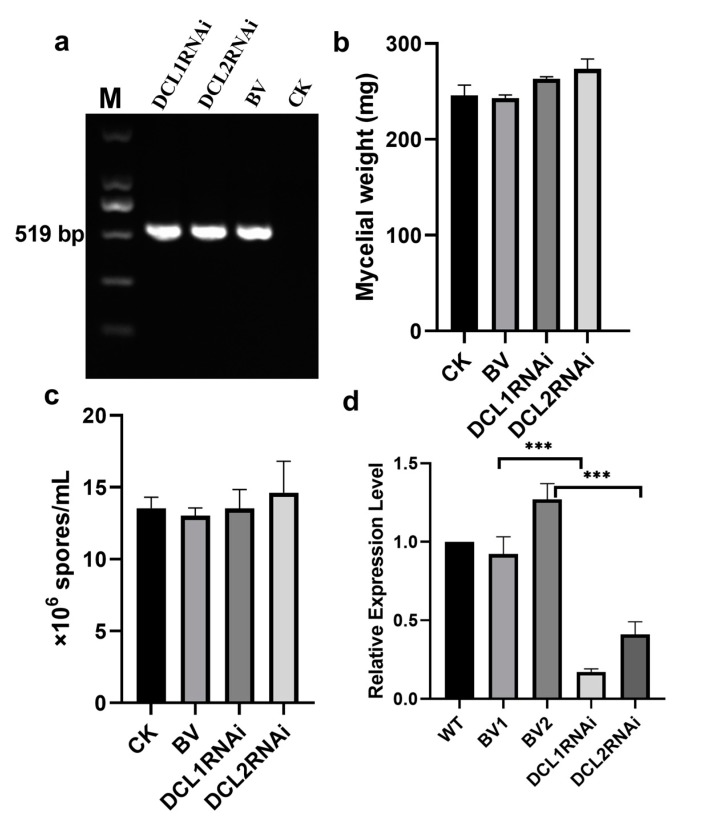
Features of the RNAi transformants. (**a**) PCR screening based on the amplification of the hygromycin resistance gene; (**b**) mycelial dry weight from wild-type *P. italicum* and the RNAi transformants; (**c**) spore number from wild-type *P. italicum* and the RNAi transformants; (**d**) qPCR expression analysis of the Dicer-like genes in the RNAi transformants and wild-type strain after culture in PDB (potato dextrose agar) for 5 days. CK (wild-type), BV (blank vector transformant), DCL1RNAi (Dicer-like1 gene transformant) and DCL2RNAi (Dicer-like 2 gene transformant), and BV1 and BV2 mean the relative expression level of Dicer-like 1 and Dicer-like 2 in the blank vector transformant, respectively. Each bar represents a mean ± standard error (*n* = 3 or 4), and the asterisk indicates a significant difference (***, *p* < 0.001).

**Figure 2 cells-09-00363-f002:**
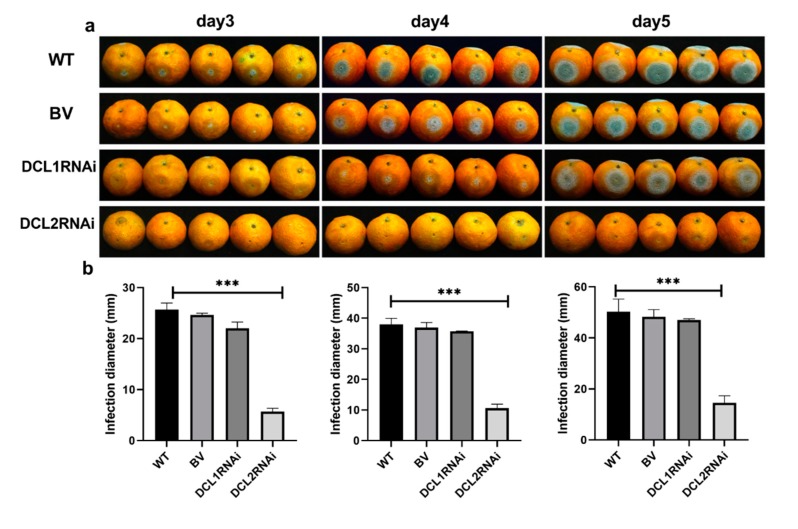
(**a**) Images of the infection of citrus fruits by *P. italicum*; (**b**) infection diameter on the surface of the citrus fruits 3, 4, and 5 days after inoculation. WT (wild-type), BV (blank vector transformant), DCL1RNAi (Dicer-like 1 gene transformant), and DCL2RNAi (Dicer-like 2 gene transformant). Each bar represents a mean ± standard error (*n* = 3), and the asterisk indicates a significant difference (***, *p* < 0.001).

**Figure 3 cells-09-00363-f003:**
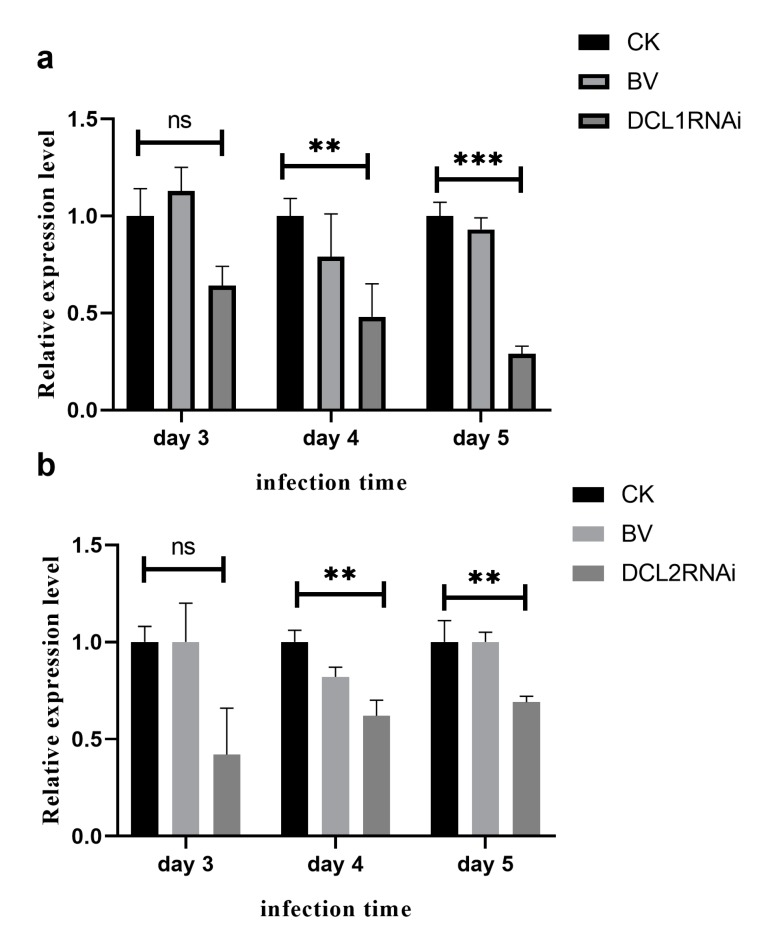
qRT-PCR expression analysis of the Dicer1-like gene (**a**) and Dicer2-like gene (**b**) in *P. italicum* after inoculation on citrus fruit epidermis for 3, 4, and 5 days. CK (wild-type), BV (blank vector transformant), DCL1RNAi (Dicer-like 1 gene transformant), and DCL2RNAi (Dicer-like 2 gene transformant). Each bar represents a mean ± standard error (*n* = 3), and the asterisk indicates a significant difference (**, *p* < 0.01, ***, *p* < 0.001), no significance (ns).

**Figure 4 cells-09-00363-f004:**
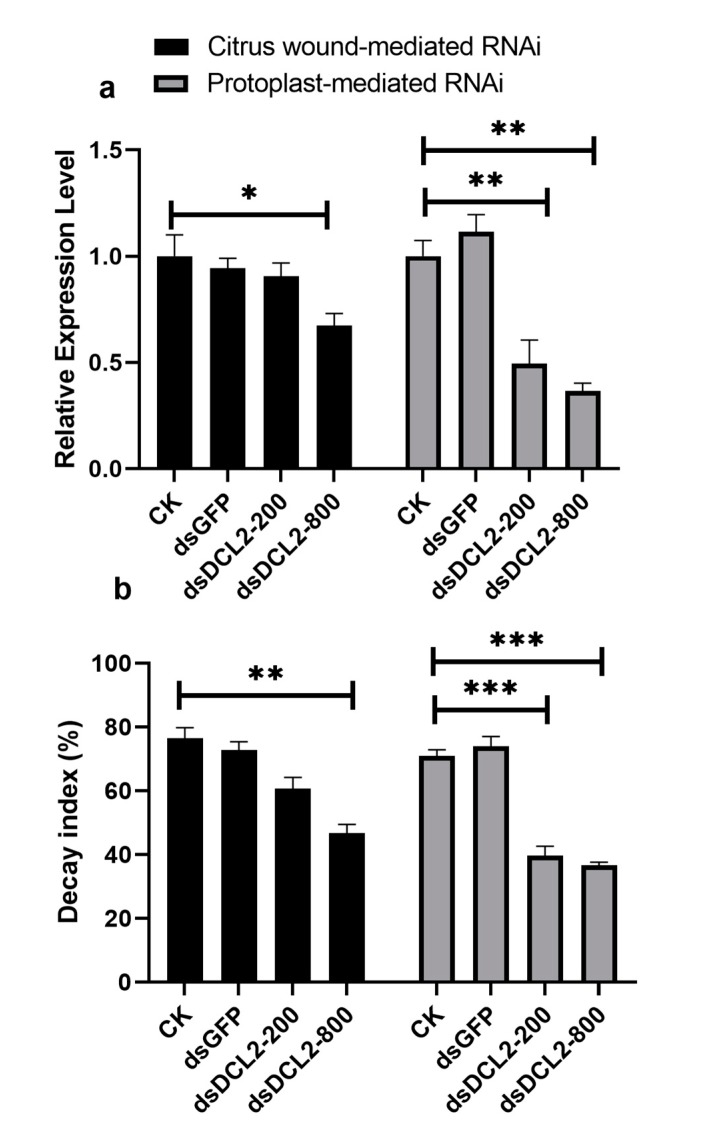
Effects of citrus wound-mediated RNAi and protoplast-mediated RNAi on the expression profile of the Pit-DCL2 gene (**a**) and the citrus decay index caused by wild-type *P. italicum* infection (**b**). dsDCL2-200 and 800 mean the concentration of the treated dsRNA is 200 ng and 800 ng, respectively. Each bar represents a mean ± standard error (*n* = 3), and the asterisk indicates a significant difference (*, *p* < 0.5, **, *p* < 0.01, ***, *p* < 0.001).

**Figure 5 cells-09-00363-f005:**
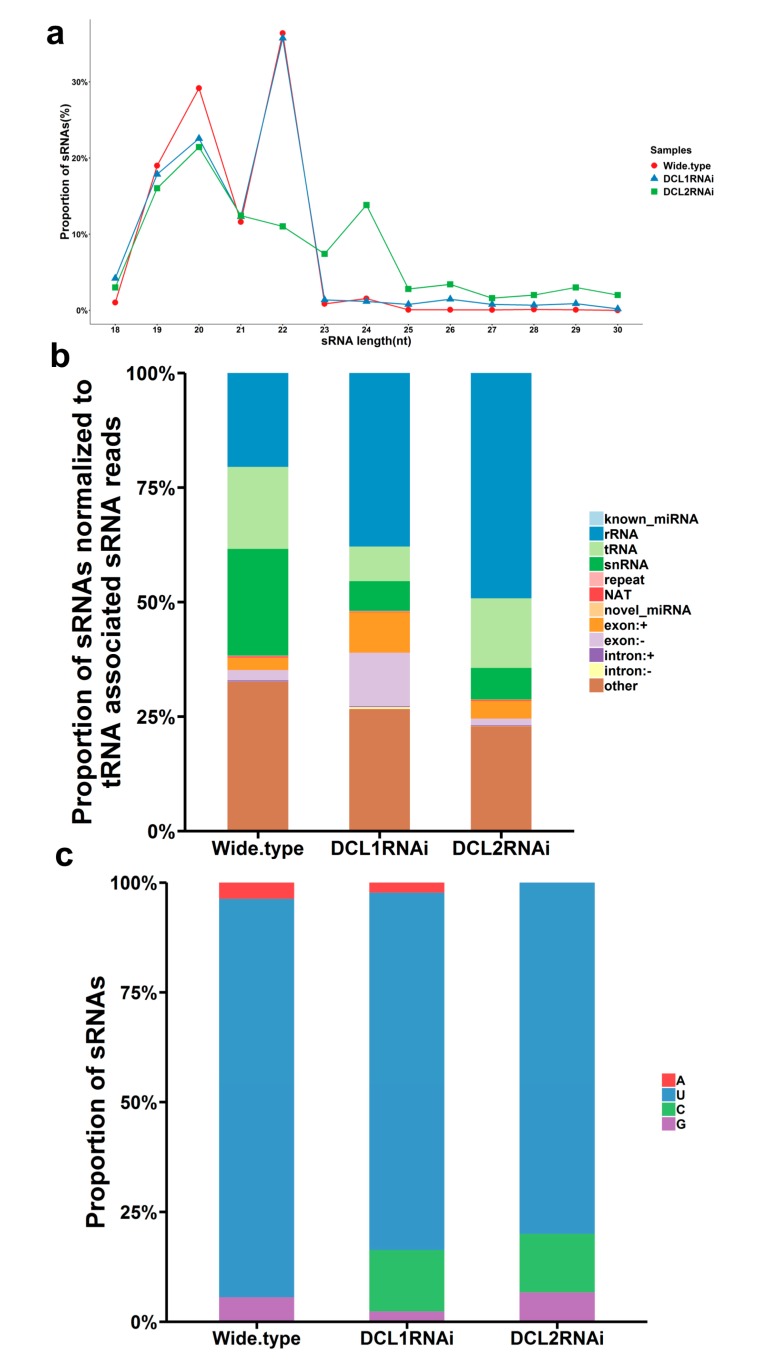
Characteristics of the small RNAs (sRNAs) in different samples. (**a**) Size distribution of sRNA from wild-type *P. italicum* and the RNAi transformants. (**b**) Proportion of total genome-matched sRNA from wild-type *P. italicum* and the RNAi transformants. (**c**) 5′ Nucleotide preference for sRNAs in wild-type *P. italicum* and the RNAi transformants.

**Figure 6 cells-09-00363-f006:**
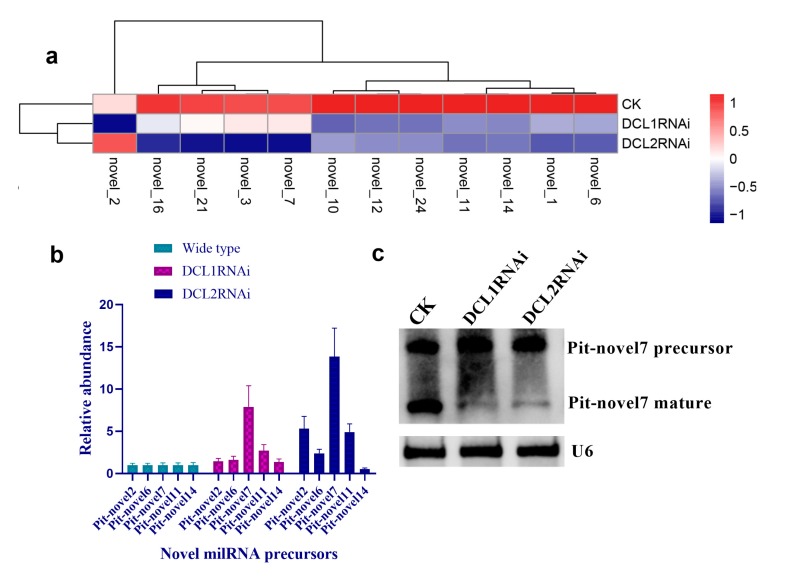
Characteristics of novel microRNA-like RNAs (milRNAs) in *P. italicum.* Heatmap of the differentially expressed milRNAs based on small RNA sequencing (**a**). RT-qPCR analysis of the precursors of the novel milRNA precursors (**b**). Northern blot analysis of Pit-novel7 milRNA, and U6 was used as an internal reference (**c**).

**Table 1 cells-09-00363-t001:** Characteristics of the predicted novel milRNAs and their targets in *Citrus sinensis.*

Mature_ID	Mature_Seq	Precursor Position	MFE	Target Accession	Target Annotation	Alignment
Pit-novel1	ugccaaaguaguuggacucgcu	contig333:14656..14796:+	−50.4	Cs7g21420 Cs7g21400	Leucine-rich repeat (LRR) receptor kinase MDIS1-interacting receptor like kinase 2	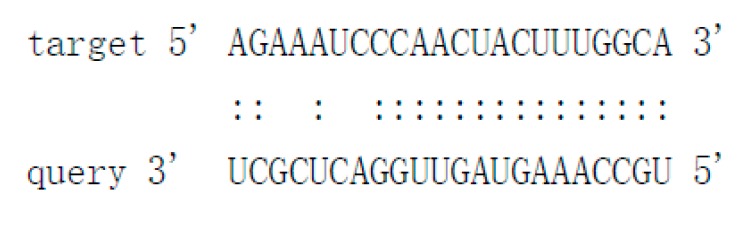 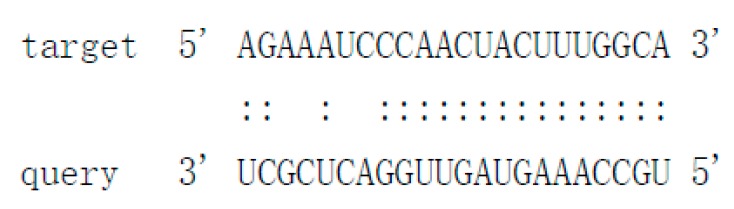
Pit-novel2	uaauugaucccgcuugcucccacg	contig2630:133598..133733:+	−75.9	N/A	N/A	
Pit-novel3	uauggugacaaaaggcuucauu	contig1137:40522..40624:-	−49.8	N/A	N/A	
Pit-novel6	uacguagcagcgauccucuagc	contig837:270..404:+	−43.7	N/A	N/A	
Pit-novel7	uggcuggagcaugcgcuugauu	contig1283:115664..115928:-	−90.6	Cs7g19510	AP2/B3-like transcriptional factor	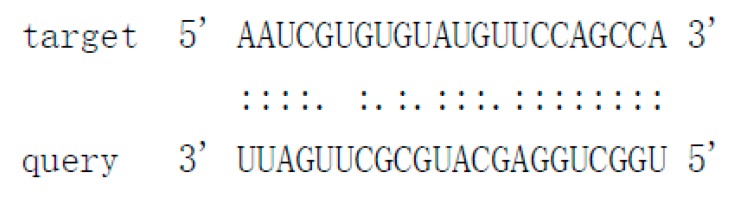
Pit-novel10	ugggaccuccgaaguauucgg	contig80:152579..152678:+	−61.14	N/A	N/A	
Pit-novel11	uuugggugauuuucaggcuc	contig988:35989..36189:+	−77.72	Cs4g07930	Subtilisin-like protease	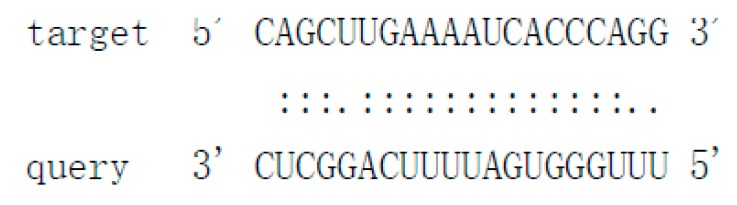
Pit-novel12	uacuucggaggucccacugu	contig2212:6256..6357:-	−64.1	N/A	N/A	
Pit-novel14	ugaguaggagagucauuugcu	contig769:62932..63030:+	−51.3	Cs7g09590	Protein-tyrosine-phosphatase MKP1-like isoform X1	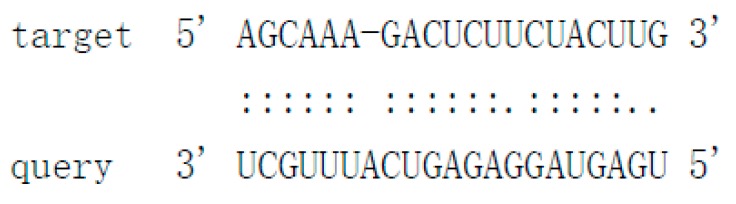
Pit-novel16	uaugacgauacugcuccauac	contig1182:333..382:+	−17.1	N/A	N/A	
Pit-novel21	uggugacaaaaggcuucauuuu	contig50:167259..167366:-	−44.9	N/A	N/A	
Pit-novel24	uacuucggaggucccacuguacga	contig840:52392..52496:+	−65.4	N/A	N/A	

Note: N/A (not applicable), MFE (minimum free energy), Pit (Penicillium italicum).

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
