# Peer review of "Silencing Dicer-Like Genes Reduces Virulence and sRNA Generation in Penicillium italicum, the Cause of Citrus Blue Mold"

_cells, 2020, doi:10.3390/cells9020363_

Round 1
Reviewer 1 Report
I already know this manuscript because I have reviewed it once. In then I had some comments which the authors have included in the current version of the manuscript. In my opinion, the current version of the manuscript looks much better than the original one. Therefore, I maintain a high rating for this work.
Author Response
I already know this manuscript because I have reviewed it once. In then I had some comments which the authors have included in the current version of the manuscript. In my opinion, the current version of the manuscript looks much better than the original one. Therefore, I maintain a high rating for this work
Response: Thank you so much for positive comments on our manuscript.
Reviewer 2 Report
As data reproducibility has become a crisis in science, there is concern that some very simple experiments are not reproduced. The materials and methods mention biological replicates, but these do not appear in the data sets or in the statistical analysis. The authors should convince us that the CCL2RNAi results are biologically reproducible in their lab and not simply technically reproducible. For example, in Figures 1 and 3, it is more likely that n=1, with 3 technical replicates. Figure 2 is a 5-day experiment, and easily reproduced.
Author Response
As data reproducibility has become a crisis in science, there is concern that some very simple experiments are not reproduced. The materials and methods mention biological replicates, but these do not appear in the data sets or in the statistical analysis. The authors should convince us that the CCL2RNAi results are biologically reproducible in their lab and not simply technically reproducible.
Response:we believe that the CCL2RNAi results are biologically reproducible, as required to response in the cover letter, please find it.
For example, in Figures 1 and 3, it is more likely that n=1, with 3 technical replicates.
Response:the biomass data showed in Figure 1b and c were performed 4 biologically repeats with 3 technical replicates for each one as described in line 172-173; the Figure 1d and Figure 3 were qPCR data,which is 3 biologically repeats with 3 technical replicates as described 253.
Figure 2 is a 5-day experiment, and easily reproduced.
Response:Figure 2 was re-done once more as your required and the details were shown in the cover letter. Because this paper was treated as a re-submission version and the response are following the cover letter.